# Data-Driven Prediction of Embryo Implantation Probability Using IVF Time-lapse Imaging

**David H. Silver**[*][1]                                              DAVID@EMBRYONICS.ME
**Martin Feder**[1]                                                 MARTIN@EMBRYONICS.ME
**Yael Gold-Zamir**[1]                                                 YAEL@EMBRYONICS.ME
**Avital L. Polsky**[1]                                              AVITAL@EMBRYONICS.ME
**Shahar Rosentraub**[1]                                            SHAHAR@EMBRYONICS.ME
**Efrat Shachor**[1]                                                 EFRAT@EMBRYONICS.ME
**Adi Weinberger**[1]                                                  ADI@EMBRYONICS.ME
**Pavlo Mazur**[2]                                                   P.MAZUR@IVF.COM.UA
**Valery D. Zukin**[2]                                               V.ZUKIN@IVF.COM.UA
**Alex M. Bronstein**[3,1]                                         BRON@CS.TECHNION.AC.IL

[1] *Embryonics, Tel Aviv, Israel*

[2] *Clinic of Reproductive Medicine 'Nadiya', Kyiv, Ukraine*

[3] *Department of Computer Science, Technion – Israel Institute of Technology, Haifa, Israel*

## Abstract

The process of fertilizing a human egg outside the body in order to help those suffering from infertility to conceive is known as *in vitro* fertilization (IVF). Despite being the most effective method of assisted reproductive technology (ART), the average success rate of IVF is a mere 20-40%. One step that is critical to the success of the procedure is selecting which embryo to transfer to the patient, a process typically conducted manually and without any universally accepted and standardized criteria. In this paper we describe a novel data-driven system trained to directly predict embryo implantation probability from embryogenesis time-lapse imaging videos. Using retrospectively collected videos from 272 embryos, we demonstrate that, when compared to an external panel of embryologists, our algorithm results in a 12% increase of positive predictive value and a 29% increase of negative predictive value.

**Keywords:** Deep Learning, In Vitro Fertilization, Embryo Selection, Video Classification.

## 1. Introduction

*In vitro* fertilization (IVF) is a procedure in which ova (egg cells) harvested from an adult female are fertilized by live sperm *in vitro*. After successful fertilization, the resulting embryos are incubated for several days while a trained embryologist manually tracks their development, using morphological and/or morphokinetic characteristics to generate a grade for each embryo indicative of its viability and likelihood of successful uterine implantation and, hopefully, live birth.

Although manual morphological annotation and quality assessment of embryos fertilized *in vitro* remains the gold standard for predicting IVF success, efforts to standardize

---

[*] Corresponding author

and improve prediction accuracy have become increasingly computational (several reviews have been published discussing such approaches from various points of view (Simopoulou et al., 2018b,a; Del Gallego et al., 2019; Liu et al., 2019; Basile et al., 2015)). Most algorithms developed for embryo outcome prediction require user-defined input parameters (such as specific morphological characteristics), execute a series of user-defined tasks, and then produce an estimated probability of achieving a user-defined outcome. Essentially, this approach can be seen as an attempt to mimic the human embryologist. While algorithms of this nature may help embryologists to more efficiently assess embryo quality, they are limited in their ability to improve outcomes as they are often dependent on the same scoring parameters as manual assessment, which is highly variable between observers (Khosravi et al., 2019; Adolfsson et al., 2018; Adolfsson and Andershed, 2018; Uyar et al., 2015; Paternot et al., 2011; Martínez-Granados et al., 2018). Lack of standardization and agreement on criteria likely contribute to the low success rate of IVF. Researchers in the assisted reproductive technology (ART) community have, therefore, increasingly turned to machine learning techniques in recent years (Simopoulou et al., 2018b; Liu et al., 2019; Curchoe and Bormann, 2019; Wang et al., 2019; Zaninovic et al., 2019).

We introduce a novel machine learning algorithm, referred to as *Ubar*, that takes time-lapse images as the input and predicts embryo implantation probability. We compared the implantation probability predictions of the algorithm to embryo grades provided by an external panel of embryologists and to the known ground truth implantation results.

## 2. Data

Our dataset consisted of 8,789 retrospectively collected time-lapse videos of developing embryos, 4,087 of which were graded by an external panel of embryologists. Of the transferred embryos with known implantation data (KID), 216 were assigned the label of successful implantation (transfers that resulted in the detection of a gestational sac and fetal heartbeat at 7 and 12 weeks gestation). 56 embryos were assigned the label of failed implantation (no detection of gestational sac).

## 3. Methods

A CNN autoencoder was trained with the $L_2$ loss on the individual frames from the unlabeled videos. The encoder comprising 10 layers was used to produce a 968-dimensional embedding per frame. An LSTM network was trained on the 4,087 graded videos receiving the embeddings of the sequence of frames and predicting the embryologist grade distribution.

The same network was used with a different binary head to predict the implantation probability on the 272 videos with known implantation data. Embryologist-graded and KID data were structured as 10 cross-validation folds, assuring no inclusion of the same patient data into training or validation sets. In order to compare UBar performance to current embryo selection standards, an external panel of five embryologists from various countries (India, Latvia, Ukraine, and the United States) assigned each embryo video a grade between 1 and 5, with 1-2 corresponding to the recommendation not to transfer due to poor embryo

quality, while 4-5 being a recommendation to transfer due to the perceived high likelihood of successful implantation.

## 4. Results

Receiver operating characteristic (ROC) curves were calculated for both UBar predictions and panel scores, with thresholds between 0 and 1 (UBar) or 1 and 5 (panel) and are depicted in Figure 1**A**. The area under the curve (AUC) of UBar was $0.82 \pm 0.07$, outperforming the expert panel (AUC $= 0.58 \pm 0.04$). Means and standard deviation for UBar were computed using bootstrapping over 1000 repetitions. In order to achieve a more clinically-relevant assessment of UBar's performance, the positive (PPV) and negative (NPV) predictive values were calculated for UBar predictions and compared to those of the expert panel grades (Figure 1**B**). PPV corresponds to the number of embryos correctly predicted as successful implantation divided by the total number of embryos predicted as successful, while NPV corresponds to the number of embryos correctly predicted as failed implantation divided by the total number of embryos predicted to fail. Both the PPV (93%) and NPV (58%) of UBar significantly exceeded the corresponding values of the expert panel ($81 \pm 1\%$ and $23 \pm 8\%$, respectively), implying that application of UBar in a clinical setting could potentially improve embryo transfer outcomes.

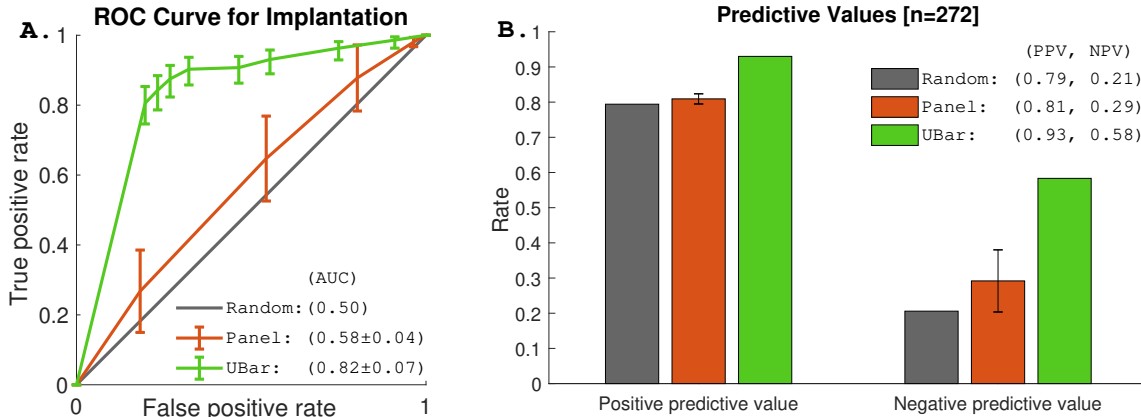

Figure 1: **A.** Performance of UBar compared to an expert panel of embryologists. **B.** Predictive values of UBar, expert panel, and a random model of which the values correlate to the prevalence of each class in the dataset: 79% successfully implanted and 21% failed implantation.

## 5. Discussion

A previously published study by (Tran et al., 2019) showed that time-lapse imaging files could be used for implantation probability prediction. However, the negatively labeled samples in Tran et al.'s study included embryos that were intentionally deselected from embryo transfer, effectively predicting a different set of outcomes: the embryologists' decisions as

well as implantation probability. Including the embryologists' decisions in the outcome prediction is arguably an easier task, as their decisions are based on designated parameters (though such parameters differ between individuals), whereas the parameters that lead to successful and failed implantation are not well understood. Furthermore, increased sample sizes of training sets have been shown to improve AUC values (Stiglic et al., 2009; Wu et al., 2018), possibly contributing to the high AUC reported by Tran et al. (0.93), whose model was trained on videos from more than 10,000 embryos.

In this paper we show that, using a small number of labeled samples, we built an embryo outcome prediction model that outperforms a panel of expert embryologists. Future directions for this model include application to a larger amount of samples originating from multiple IVF clinics. Additionally, in an effort to further improve results, we are exploring variants of the neural network, such as: inclusion of additional clinical data or training the network as a whole (multi-task network training of both the auto-encoder and the classifier).

## Acknowledgments

This work was partially supported by the Israel Innovation Authority, grant #65201. We thank D. E. Fordham for critical reading of the manuscript, and A. Gershenfeld for assistance with organizing the data.

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
