# OpenReview forum: "Data-Driven Prediction of Embryo Implantation Probability Using IVF Time-lapse Imaging"
_MIDL.io/2020/Conference — MIDL 2020_

### Official Review · AnonReviewer1 · 2020-03-09
**Good work, profund evaluation, great scientific embedding**

**Rating:** 4
**Confidence:** 3

**Review:**

In this work, the authors present an LSTM-based model to predict the probability of successful embryo implantation based on time-lapse images of the developing embryo in an in-vivo environment. The results are very promising and discussed very well, the work is eagerly embedded within the field.
The overall impression of the paper is very good. The scientific background is profund and the description of the methods is clear. One minor concern is the description of the data set used, as I got a bit confused by the 8,789 data points of which only a subset is used as only these were labeled. Additionally, the presence of two different sets of expert panels was a bit confusing as well. These parts could be explained more clearly.
However, the authors put a lot of effort into creating a sufficient data set with a comparison to the clinical state-of-the-art and hence deliver a very good and reliable study. I recommend to accept this paper for MIDL 2020 and am sure that there will be interesting discussions during the conference!

---

> ### Public Comment · ~Embryonics_Review1 · 2020-05-25
> **Thank you for reading our manuscript and for your remarks.**
>
> We will clarify how the unlabelled videos were used:
> Because labels are unnecessary in the stage of the encoder, which converts a 2d image frame to a 1d vector that should faithfully represent the original image, we can use any image of a developing embryo (including those derived from unlabelled videos). During training, a network (“autoencoder”) embeds the image from an unlabeled dataset into a 1d vector, and then another network converts the 1d vector back to the original 2d image - allowing us to confirm the existence of the information in the 1d embedding.
>
> So the 8,789 samples, of which only a small subset was labelled, was used for training the autoencoder.

---

### Official Review · AnonReviewer3 · 2020-03-09
**Good results but lack of more technical details.**

**Rating:** 2
**Confidence:** 4

**Review:**

The authors present a CNN+LSTM network architecture denoted as UBar for the prediction of embryo implantation in IVF. The main task to solve with the machine learning algorithm consists of analyzing a time series of images to make a binary prediction.

The results presented look good when comparing to other published works. However, I find the content of the article poor in terms of motivation to use this specific architecture. The authors are focused on the limitations of using user-defined parameters while there are already some works that use machine learning methods. Hence, I would strongly recommend them to include further information in this direction that supports their work.

The architecture is defined as "UBar" but it makes no sense to me due to the lack of details. I would recommend them to include a further enough description of the network so any reader can understand what they did. Additionally, the code for the work or a comparison with other methods is not available, which makes all of it difficult to reproduce or evaluate the reported values.

---

> ### Public Comment · ~Embryonics_Review1 · 2020-05-25
> **We thank the reviewer for reading the manuscript, commenting, and expressing confidence in our results.**
>
> In a short paper there is no revision period, as well as a tight page limit that we exhausted. We are considering if there is a way to release the code and some material in an appropriate system after considering issues related to medical confidentiality and intellectual property.

---

### Official Review · AnonReviewer2 · 2020-03-12
**Exciting results on a well-motivated problem**

**Rating:** 4
**Confidence:** 3

**Review:**

The authors present a deep learning method for predicting embryo implantation probability, based on time-lapse videos acquired during IVF. The authors claim a substantial improvement relative to an expert panel of embryologists, as measured by AUC and predictive values. The method is assessed using 10-fold cross validation on 272 videos with known implantation outcome, and 4,087 videos with panel grading.

There is some pre-existing similar work in the literature - most similarly Tran et al 2019. This is appropriately cited by the authors, who describe subtle differences compared with their work. Despite this similarity, replication of a solution to the general problem on different data (using the authors' method rather than that of Tran et al) counts as sufficient originality. Significance of the work seems high: the problem is clearly important, and the potential for improvement over current clinical methods seems substantial. Quality of the work seems high, particularly given the short format: the authors present a convincing method and then validate it quite thoroughly. Clarity is good, although it would be beneficial to introduce the work of Tran et al earlier, and better explain the differences in the authors' work. The authors also do not explain their method in great detail, although I feel this is understandable given the short format.

On balance I am impressed by this paper, and strongly believe it should be accepted.

Pros:
* Well-motivated.
* Well-described.
* High quality validation.

Cons:
* Fairly similar to pre-existing work, although I think this is perfectly fair and the work still has significant originality.
* Slight lack of clarity in differences vs previous work.
* Lack of detail concerning the authors' method.
* Confusing description of the dataset: how were the non-labelled videos used?

---

> ### Public Comment · ~Embryonics_Review1 · 2020-05-25
> **We thank the reviewer for taking the time to read and evaluate our manuscript.**
>
> We agree that some clarification would have been better, and appreciate the reviewer’s  understanding of the tight 3 page limit on short papers.
>
> We will clarify how the unlabelled videos were used:
> Because labels are unnecessary in the stage of the encoder, which converts a 2d image frame to a 1d vector that should faithfully represent the original image, we can use any image of a developing embryo (including those derived from unlabelled videos). During training, a network (“autoencoder”) embeds the image from an unlabeled dataset into a 1d vector, and then another network converts the 1d vector back to the original 2d image - allowing us to confirm the existence of the information in the 1d embedding.

---

### Official Review · AnonReviewer4 · 2020-03-13
**Good evaluation and promising results, with missing details and poor discussion**

**Rating:** 3
**Confidence:** 3

**Review:**

Overall, the quality of the paper is fair. It is well-written, well-structured and easy to read for someone without knowledge on IVF and ART. The method is compared to five embryologists and results clearly shows that learning directly from the clinical outcome outperfoms embryologists by a large margin. The main weakness of the paper is in the methods section.

The methodological novelty seems insignificant. Plenty of works combine autoencoders with LSTMs. I suggest you either argue for the novelty or remove the claim from the paper.

The methods section lacks details for reproducing the work. These must be provided in a supplement to allow reproducability. If you want your work applied in clinics, this is much more important than improving the results.

In the methods section you describe training an autoencoder on unlabeled data, then training an LSTM using autoencoder embeding and embryologist grades. As I read it, UBar is the same LSTM just trained on clinical outcomes. You do not report results for the embryologist trained LSTM, so what do you use this LSTM for? If you dont use it, remove it from the section. If you do use it, you cannot argue that you learn from "a small number of labeled samples" as done in the final paragraph of the paper.

In the discussion you almost exclusively focus on the work by Tran et al and why comparing with that work is unfair. Instead, you should have made the comparison and highlighted the differences clearly. What is interesting is not who is better, but how, and how well, the task can be solved.

You argue that including embryologists decisions in the prediction is an easier task. I am not convinced. In your case, you train on data that has already been filtered to only include positive decisions by embryologists, otherwise the eggs would not have been implanted. It is not obvious how to best get around this issue, since the first embryologist screening probably has false negatives, but you need to take it into account.

Your statement about AUCs and training sizes is either obviously correct or obviously wrong, depending on interpretation. The only way training size can influence AUC is by influencing the training of the model. It is quite well known that more training data, in general, results in improved performance of networks. This holds for all the popular performance measures. Having said that, if the model predictions does not change, then AUC does not change. Maybe you meant the size of the test set? In that case, it is the ratio of positive/negative that is relevant. Regardless, trying to paint others work negatively by arguments to some general issue with established performance metrics is disingenuous. If there is an issue with Tran et al you should state it clearly, if not, you should accept their results.

A mior nitpick:
You define all abbreviations except for UBar. It is fine that you give your method a name (although I personally dislike it), but a bit weird not to explain it.

Finally, I would very much have liked to to see a frame from one of the videos. I am aware of the page limitation, so maybe MIDL should allow an extra page solely for an image of the raw data.

---

> ### Public Comment · ~Embryonics_Review1 · 2020-05-25
> **Thank you for your detailed examination of our work and for the constructive criticism.**
>
> We would like to point out that this is a short paper, with a 3 page limit, so most of the brevity comes from that limitation. We agree that an in-depth comparison of the network performance with grading based training alone would be enlightening, however we attempted  to include as much information about both the methods and the results as possible without exceeding  the allotted limit. We will soon publish a more extended version of the paper covering these details.
>
> Similarly, this was not a paper about general methods in classification metrics. So we, maybe wrongly, assumed that our short claim about the weaknesses of measuring AUC in a selection algorithm would be enough. Regarding the hypothetical improvement, if we had trained on 30 times that data, is an hypothetical claim which has a basis from how we know neural networks to behave. We should have shown how the AUC improves specifically on our data when we increase the training set and extrapolate a better informed number of a comparable AUC to Tran et al. Naturally, it would mean new predictions. We still hold the opinion that AUC is less meaningful for clinical applications than PPV & NPV, both because it is hard to estimate the clinical efficacy and due to the impossibility to calculate a smooth RoC curve for an embryologist grading.
>
> As for the novelty of the network, we agree that the novelty is less on the technical arrangement of the network, at least with the sparse details we have shared in the paper. The novelty of this model is its application to medical imaging and the three-stage training process.
>
> As for the term UBar, we are at fault. It should be “Ubar,” which is a transliteration of the Hebrew word for “embryo”: עוּבָּר.
>
> We would also like to clarify that there is no revision option for short papers, but we will make available a revised arxiv version.

---

### Meta-Review · Program_Chairs · 2020-04-10
**MetaReview of Paper126 by AreaChair1**

**Rating:** 4

**Metareview:**

There is consensus that this paper has exciting results and would generate good discussion at the conference.

**Paper Type:**

both

---

### Decision · Program_Chairs · 2020-04-11

Accept